# Prophylactic Glatiramer Acetate Treatment Positively Attenuates Spontaneous Opticospinal Encephalomyelitis

**DOI:** 10.3390/cells12040542

**Published:** 2023-02-08

**Authors:** Ümmügülsüm Koc, Steffen Haupeltshofer, Katharina Klöster, Seray Demir, Ralf Gold, Simon Faissner

**Affiliations:** Department of Neurology, St. Josef-Hospital, Ruhr-University Bochum, Gudrunstr. 56, 44791 Bochum, Germany

**Keywords:** OSE, immunomodulation, chronic inflammatory CNS disorders, progressive multiple sclerosis, NMOSD

## Abstract

**Background:** Glatiramer acetate (GA) is a well-established treatment option for patients with clinically isolated syndrome and relapsing–remitting multiple sclerosis (MS) with few side effects. The double transgenic mouse model spontaneous opticospinal encephalomyelitis (OSE), based on recombinant myelin oligodendrocyte glycoprotein_35-55_ reactive T and B cells, mimicks features of chronic inflammation and degeneration in MS and related disorders. Here, we investigated the effects of prophylactic GA treatment on the clinical course, histological alterations and peripheral immune cells in OSE. **Objective:** To investigate the effects of prophylactic glatiramer acetate (GA) treatment in a mouse model of spontaneous opticospinal encephalomyelitis (OSE). **Methods:** OSE mice with a postnatal age of 21 to 28 days without signs of encephalomyelitis were treated once daily either with 150 µg GA or vehicle intraperitoneally (i. p.). The animals were scored daily regarding clinical signs and weight. The animals were sacrificed after 30 days of treatment or after having reached a score of 7.0 due to animal care guidelines. We performed immunohistochemistry of spinal cord sections and flow cytometry analysis of immune cells. **Results:** Preventive treatment with 150 µg GA i. p. once daily significantly reduced clinical disease progression with a mean score of 3.9 ± 1.0 compared to 6.2 ± 0.7 in control animals (*p* < 0.01) after 30 d in accordance with positive effects on weight (*p* < 0.001). The immunohistochemistry showed that general inflammation, demyelination or CD11c^+^ dendritic cell infiltration did not differ. There was, however, a modest reduction of the Iba1^+^ area (*p* < 0.05) and F4/80^+^ area upon GA treatment (*p* < 0.05). The immune cell composition of secondary lymphoid organs showed a trend towards an upregulation of regulatory T cells, which lacked significance. **Conclusions:** Preventive treatment with GA reduces disease progression in OSE in line with modest effects on microglia/macrophages. Due to the lack of established prophylactic treatment options for chronic autoimmune diseases with a high risk of disability, our study could provide valuable indications for translational medicine.

## 1. Introduction

Chronic autoimmune inflammatory diseases of the central nervous system (CNS) are variable in their pathophysiology, clinical manifestation and progression as well as treatment options. The most important chronic inflammatory CNS disease is multiple sclerosis (MS), a multifactorial widespread inflammatory CNS disease that manifests in the majority of the patients with a relapsing–remitting phenotype (RRMS) [1]. After, on average, 15–20 years without treatment, the disease transforms into a secondary progressive phenotype, characterized by a gradual increase in neurological impairment [2]. A minority of 10–15% of patients present initially with a primary progressive disease. In contrast to MS, neuromyelitis optica spectrum disorder (NMOSD) is a mostly antibody-associated chronic autoimmune inflammatory CNS disorder [3]. While modern immunomodulatory therapeutics are highly effective in reducing relapse activity, disability progression is often less successfully stopped [4]. Thus, investigating the potential of therapies to slow down disease progression in chronic inflammatory CNS disorders is of high interest.

One medication used for the treatment of chronic inflammatory CNS disorders is glatiramer acetate (GA), a well-tolerated injectable immunomodulatory medication consisting of four different amino acids, which has a good safety profile and is authorized for clinically isolated syndrome and RRMS [5,6]. Although this drug has been used for more than two decades in MS and a plethora of mechanisms have been described, its precise mechanism of action is incompletely understood [7]. To further elucidate whether GA might also mitigate effects on progression, we took advantage of a spontaneous model of multiple sclerosis, opticospinal encephalomyelitis (OSE). OSE represents an animal model mimicking features of both MS and NMOSD, such as spontaneous onset and continuous progression. Mating mouse strains with a transgenic myelin oligodendrocyte glycoprotein (MOG)_35–55_ reactive T cell (2D2) and B cell receptor (TH), double transgenic mice are bred, which develop spontaneous autoimmune neuroinflammation [8]. Here, we investigated prophylactic GA treatment in OSE mice regarding effects on disability progression as well as effects on histological alterations and lymphocyte subsets. We show that prophylactic GA treatment in OSE slightly but significantly improves clinical progression in accordance with modest effects on microglia/macrophages. 

## 2. Methods

### 2.1. Animals

All animal experiments were approved by and performed in compliance with North-Rhine–Westphalia animal welfare regulations (84-02.04.2015.A041). To induce spontaneous opticospinal encephalomyelitis (OSE), we used double transgenic mice with recombinant T and B cells for myelin oligodendrocyte glycoprotein MOG_35–55_ antigen as previously described [9,10]. The transgenic mouse strains TCR^MOG^ (2D2) [11] and IgH^MOG^ (TH) [12] were bred and housed in the animal facility of Ruhr-University Bochum. OSE mice develop spontaneous clinical symptoms within approximately 28 days [9]. At a postnatal age of 21 to 28 days, OSE mice without clinical signs were randomized into two groups and treated once daily either with 150 µg glatiramer acetate (GA) (n = 18) or 150 µL phosphate buffered saline (PBS) as vehicle (n = 20) intraperitoneally (i. p.). Mice were weighed and scored daily regarding clinical signs (scoring from 0 = no symptoms, 1 = reduced tone of tail, 2 = limp tail, 3 = absent righting, 4 = gait ataxia, 5 = mild paraparesis of hindlimbs, 6 = moderate paraparesis, 7 = severe paraparesis/paraplegia, 8 = tetraparesis, 9 = moribund, 10 = death) [9]. Animals were sacrificed after a treatment period of 30 days or when they reached a score of 7.0 due to animal care guidelines. 

### 2.2. Flow Cytometry

Animals were sacrificed, and secondary lymphoid organs were isolated. Splenocytes and lymph node cells were harvested and washed with PBS before cell cycle analysis using propidium iodide staining (50 µg/mL) was performed. Surface protein staining was conducted using incubation with fluorescent-labeled antibodies against targets, as listed in Appendix A, at 4 °C for 40 min. For intracellular staining, cells were fixed in 4% paraformaldehyde (PFA) at 4 °C for at least 30 min, permeabilized in permeabilization buffer (BD Biosciences) and incubated with fluorescence-marked antibodies (Appendix A) overnight at 4 °C. Cells were analyzed using a FACSCalibur™ (BD Biosciences). Lymphocytes were gated using the software CellQuest™ (BD Biosciences), followed by gaiting for respective antibodies.

The gating strategy was as follows: CD4^+^ and CD8^+^ cells were differentiated from the lymphocyte population, and CD69^+^ cells were gated from the populations above. Th1 and Th17 cells were gated from CD4^+^ cells. Regulatory T cells (Tregs) were identified following gating of CD4^+^ and CD25^+^ cells. To identify B cells, lymphocytes were gated first, followed by CD45^+^/B220^+^ cells; the population of MHCII^+^ cells was gated thereafter. The macrophages and dendritic cells populations were gated from the monocytes. MHCII^+^ cells were differentiated from these cells, depending on the cells of interest.

### 2.3. Histological Analyses

Mice were anesthetized via ketamine/xylazine injection i. p. before intracardial PBS perfusion was performed. Spinal cords were removed and fixed in 4% PFA for 24 h. Afterwards, specimens were transferred into 30% sucrose in *Aqua destillata,* followed by subdivision into cervical, thoracic and lumbar segments. All samples were snap frozen and cut axially from rostral to caudal in sections of 10 to 20 µm thickness. All sections were blinded and stained either with hematoxylin/eosin to visualize cell infiltration or immunofluorescence staining. 

### 2.4. Immunocytochemistry and Microscopy 

Immunofluorescence staining was performed on blinded sections. Blocking solution was introduced for 1.5 to 2 h, followed by incubation with primary antibodies overnight at 4 °C. FluoroMyelin Red (Appendix A) staining was performed to investigate demyelination. Microglia were identified via anti-Iba1 and anti-F4/80 staining, and dendritic cells were stained using anti-CD11c (Antibody information Appendix A). Cell nuclei were counterstained with DAPI Fluoromount-G^®^. Images were taken either using phase contrast microscopy (Olympus BX51) through a 4x objective lens or immunofluorescent microscopy (Olympus XM10, Zeiss Microscope Zen and Zeiss Axio Observer) through a 10x objective lens. Images of HE stains and FluoroMyelin Red stains were merged via Image Composite Editor (Microsoft Corporation, Redmond, WA, USA). Merging of Iba1-, F4/80- and CD11c-stains was performed automatically via Zeiss Zen Blue Software. All images were blinded prior analysis with AntRenamer (Antoine Potten). The fluorescence intensity area of Iba1-, F4/80 and CD11c was measured using the “ImageJ” software program. First, the white matter was labeled in all images. Thereafter, the image was converted to black and white (8 bit), and the threshold was measured. The intensity measurement of the entire white matter area was analyzed using a macro. General infiltration and demyelination were calculated as % of white matter.

### 2.5. Statistical Analysis

Statistical analysis was performed using GraphPad Prism software version 9 (La Jolla, CA, USA). Animal scores, weight and histological data were analyzed with non-parametric two-tailed Mann–Whitney *t*-test. For in-vitro analysis, *t*-test was used. A *p*-value *p* < 0.05 (*) was considered as statistically significant. Data are presented as mean ± standard error of mean (SEM).

## 3. Results

### 3.1. Prophylactic GA Treatment Improves Clinical Course and Body Weight in a Spontaneous Model of Autoimmune Encephalomyelitis 

To investigate whether GA treatment might elicit positive longitudinal effects on the course of opticospinal encephalomyelitis (OSE), OSE mice were treated once daily before symptom onset, either with 150 µL of GA (1 µg/µL) i. p. at a postnatal age of 24.7 ± 0.6 d (mean ± SEM) or the equivalent amount of vehicle i. p. at a postnatal age of 24.9 ± 0.6 d (n = 18 in the GA and n = 20 in the vehicle group) (Figure 1a). The incidence of OSE was 95% in the control group compared to 67% in the GA group (*p* < 0.05; Figure 1b; Table 1). To understand whether sex might influence the course of OSE, we analyzed OSE incidence depending on sex. The incidence in the vehicle group was 100% in males compared to 83% in females (*p* < 0.05). The incidence in the GA group was 57% in males compared to 100% in females (*p* < 0.05). Thus, there was no obvious incidence pattern depending on sex (Figure 1c). Vehicle-treated animals gradually progressed, as expected, and reached a score of 6.2 ± 0.7 (mean ± SEM) after an observation period of 30 d. GA-treated animals, on the contrary, had significantly less disability progression with a score of 4.1 ± 1.0 at the end of the experiment (*p* < 0.01) (Figure 1d). Again, we analyzed the influence of sex on disability progression. Interestingly, male control mice had significantly higher scores over the course of the experiment compared to females (*p* < 0.001; Figure 1e). In the GA group, female mice had higher scores over the course of the experiment (*p* < 0.05), with nearly the same end score at the end of the experiment. Positive effects on the disability progression of GA were also reflected in a higher body weight in GA-treated animals with an average of 14.4 ± 1.0 g compared to 11.1 ± 0.9 g on the last day of the experiment (*p* < 0.001; Figure 1g). Survival, however, did not differ between the groups (Figure 1h).

### 3.2. Prophylactic GA Treatment Slightly Reduces Overall Microglia/Macrophage Infiltration 

We then performed histological analyses of spinal cord sections to investigate general inflammation, demyelination and infiltration of lymphocyte subsets. First, we investigated general inflammation with H&E staining of the white matter of the spinal cord. Vehicle-treated animals had an infiltration ranging from 9.4 ± 4.2% in cervical to 16.3 ± 6.7% in lumbar sections. The mean overall infiltration of all spinal cord areas combined was 13.4 ± 3.5% (Figure 2a–c). While GA treatment trended towards less infiltration in cervical cord regions, there was a trend towards more infiltration in the thoracic and lumbar region with an overall infiltration of 13.4 ± 4.2%, which did not differ compared to the control group. Analysis of demyelinated areas of the white matter of the spinal cord showed conclusively no effects of GA treatment on demyelination. In the vehicle group, demyelination was 13.9 ± 3.4% compared to 15.9 ± 4.0% in the GA group. We then analyzed spinal cord sections regarding immune cell infiltration. Analysis of the CD11c^+^ area representing dendritic cells showed negligible infiltration both in the control group (0.4 ± 0.1%) and the GA group (0.4 ± 0.1%) (Figure 3a–c). The Iba1^+^ area, representing microglia/macrophage infiltration, showed a slight but significant decrease in GA-treated mice (11.2 ± 3.7) compared to the control group (12.9 ± 2.4; *p* < 0.05) (Figure 3d–f). Activated macrophages, identified using F4/80 staining, were also reduced following GA treatment (*p* < 0.01; Figure 3g–i). 

### 3.3. Prophylactic GA Treatment Differentially Affects Lymphocyte Subpopulations 

To better understand the effects of GA on immune cells in the OSE model, we performed analyses of lymphocyte subsets in secondary lymphatic organs at the end of the experiment. CD4^+^ T helper (spleen: control 33.5 ± 4.2%, GA 32.9 ± 3.8%; LN: control 63.7 ± 4.9%, GA 57.5 ± 5.1%) and CD8^+^ cytotoxic T cells (spleen: control 3.4 ± 0.3%, GA 4.6 ± 0.8%; LN: control 2.3 ± 0.5%, GA 4.5 ± 1.1%), as well as activated CD4^+^ T cells (CD4^+^CD69^+^ spleen: control 7.7 ± 1.7%, GA 8.7 ± 0.7%; LN: control 11.6 ± 2.8%, GA 13.8 ± 2.0% and CD8^+^CD69^+^ spleen: control 42.4 ± 5.2%, GA 41.0 ± 7.0%; LN: control 48.7 ± 4.8%, GA 46.2 ± 11.5%), showed similar frequencies in the spleen and the lymph nodes in both groups (Figure 4a,b). T helper subsets, such as CD4^+^ IFN-γ^+^ Th1 cells (spleen: control 2.7 ± 1.0%, GA 3.6 ± 1.2%; LN: control 3.0 ± 1.2%, GA 3.0 ± 0.7%) and CD4^+^IL-17^+^ Th17 cells (spleen: control 0.3 ± 0.1%, GA 0.5 ± 0.1%; LN: control 0.6 ± 0.2%, GA 0.4 ± 0.2%), were also unchanged (Figure 4c,d).

We could, however, document a positive trend for regulatory T cells (spleen: control 6.7 ± 2.0 %, GA 12.2 ± 3.7%; LN: control 4.1 ± 0.6%, GA 6.7 ± 1.4%) in GA-treated animals, which lacked significance (Figure 4e). B-cell frequencies of un-activated (spleen: control 30.1 ± 3.1%, GA 33.2 ± 3.8 %; LN: control 18.0 ± 3.7%, GA 22.1 ± 3.9 %) or activated cells (spleen: control 92.8 ± 2.2%, GA 94.9 ± 1.4%; LN: control 94.1 ± 2.4%, GA 95.6 ± 2.1%) did not differ between both groups (Figure 4f). 

Unactivated (spleen: control 5.0 ± 0.7%, GA 4.1 ± 0.5%, LN: control 2.6 ± 0.6%, GA 1.5 ± 0.4%) and activated (spleen: control 49.3 ± 7.6%, GA 32.4 ± 5.7%; LN: control 73.1 ± 7.3%, GA 71.0 ± 9.1%) CD11b^+^ MHCII^+^ macrophages, as well as unactivated dendritic cell (DC, spleen: control 11.6 ± 1.2%, GA 10.4 ± 1.7%; LN: control 4.7 ± 1.1%, GA 4.1 ± 1.3%) frequencies, revealed similar findings in GA-treated and untreated mice. However, activated (CD11c^+^ MHCII^+^) DCs were significantly increased in GA-treated mice compared to the control group both in the spleen (control 28.9 ± 5.2%, GA 46.7 ± 3.4%; *p* = 0.0308) and lymph nodes (control 20.7 ± 3.0%, GA 36.2 ± 5.9%; *p* = 0.0203) (Figure 4g,h).

## 4. Discussion

Advances in the understanding of chronic inflammatory diseases of the CNS have led to substantial improvements in the treatment of diseases such as MS or NMOSD, leading to efficient reduction of both relapse activity and disability progression. While MS relapses usually remit, NMOSD relapses are accompanied by massive tissue destruction with neurological sequelae. Hence, it remains crucial to prevent disability progression, especially in highly active MS and NMOSD patients. The addition of a mild immunomodulatory medication with a well-known safety profile could potentially support therapeutic effectiveness with limited side effects. 

We, therefore, investigated treatment with GA in a spontaneous model of chronic CNS inflammation with opticospinal disease manifestation, mimicking features of both MS and NMOSD. OSE mice are double transgenic for MOG-reactive T and B cells, which drive spontaneous autoimmune neuroinflammation [8]. OSE mice are also characterized by early and progressive degeneration of the visual system with impairment of retinal function, associated with OCT thinning and loss of retinal ganglion cells within six weeks after birth, as shown by our group [10], hence mimicking the neurodegenerative aspects of CNS neuroinflammation. Treatment with GA led to a lower incidence of disease activity and exhibited mild but significant effects on disability progression. Histologically, this was reflected in a reduction in macrophage/microglia infiltration, while the general infiltration of immune cells and demyelination was not affected. This contrasts with a study having shown no effect of GA on the EAE score in OSE mice [13]. One explanation for this difference might be a higher incidence of OSE documented in our experimental setting. While we observed an incidence of 56% in GA-treated mice and 90% in the control group, Bittner et al. reported an incidence of 33% in GA mice compared to 53% in control mice [13]. It could, therefore, be assumed that the colony treated in our center was more affected with more animals having developed disease, thus leading to a situation in which the effect of a therapeutic intervention would have been more pronounced. 

GA has been used for more than two decades, and a number of different mechanisms have been described, altogether positively attenuating MS, influencing both T cells, B cells and antigen-presenting cells [7]. One of the first mechanisms observed is the immunomodulation of T cells. GA induces a selected subgroup of GA-reactive T cells in the cerebrospinal fluid which are not found in the blood, thus eliciting anti-inflammatory effects [14]. Moreover, GA directs T cells to a regulatory anti-inflammatory phenotype, also reflected in the lymphocyte subpopulation analysis of the data presented here. Haas et al. showed that GA treatment in human leads to a reconstitution and increase in the total numbers of regulatory T cells [15]. Another subpopulation essentially involved in the mechanism of action of GA is CD8^+^ T cells. CD8^+^ T cells suppress pathogenic CD4^+^ T cell responses and induce myeloid cells, activating regulatory T cells in an antigen-dependent manner [16]. The adoptive transfer of GA-specific CD8^+^ T cells in EAE led to disease amelioration, supporting the essential role of those cells for immunomodulation and as mechanism of GA [16]. This is in line with findings from our study, showing a trend towards an upregulation of CD8^+^ T cells in OSE upon GA treatment. In addition, GA affects B cells. In experimental autoimmune encephalomyelitis (EAE), GA positively regulates inflammation by increasing anti-inflammatory IL-4, IL-10 and IL-13, while the pro-inflammatory cytokines IL-6, IL-12 and TNF-α are reduced [17]. Moreover, GA leads to a down-regulation of B cell-activating factor (BAFF) of the TNF family, a proliferation-inducing ligand (APRIL) and the BAFF receptor [17]. In addition to effects elicited on the lymphocyte function, GA also has potent effects on different subtypes of antigen-presenting cells. Monocytes are affected by GA, as shown in vitro, regarding reactivity after stimulation with ligands for different Toll-like receptors and in vivo with monocytes from GA-treated patients [18]. GA induces anti-inflammatory M2 monocytes, characterized by increased IL-10 and TGF-β signaling and decreased production of IL-12 and TNF-α, associated with reduced STAT-1 signaling [19]. M2 monocytes direct the differentiation of regulatory T cells. Another study found that this process is mediated by an inhibition of the type I IFN pathway in M2 polarization independently of MyD88 [20]. Macrophages are also affected by GA, leading to increased secretion of anti-inflammatory IL-10 and reduced release of TNF-α [21]. Of interest, bone-marrow-derived dendritic cells also show increased release of IL-10 without effects on TNF-α secretion [21]. Here, we reported an increase in activated DCs in secondary lymphatic organs as a possible immunomodulatory mechanism to slow down disease progression in OSE. There are also data about the positive effects of GA on microglia as resident immune cells of the CNS. GA reduces the cytokine production in T cell microglia co-cultures and attenuates the morphological transformation of ramified microglia into an activated ameboid form [22]. GA also promotes phagocytosis of microglia in vitro, suggesting beneficial effects on the removal of cellular debris [23]. Altogether, these data speak for a complex modulation of both adaptive and innate immunity mediated by GA. 

Clinical data of GA in different neuroinflammatory conditions with differing pathomechanisms are controversial. While various therapeutics exist for RRMS, few treatment options are effective in progressive forms of MS [2,4] and other chronic inflammatory diseases of the CNS, such as antibody-positive or seronegative NMOSD, although the therapeutic landscape has advanced during recent years. Indeed, treatment with GA or interferons is associated with a delayed conversion to secondary progression in MS when treatment is initiated early during the course of the disease [24]. In this retrospective study, it could, however, also be established that highly active medications such as fingolimod, natalizumab or alemtuzumab are associated with a significantly lower cumulative risk of conversion to secondary progression than GA or interferons [24]. GA also has been used in patients with NMOSD or anti-MOG positive patients. In a multicenter retrospective study on anti-MOG positive patients, prophylactic long-term therapy with GA could be defined as a striking feature to control disease progression [25]. A small number of patients had been treated with GA, leading to inconsistent effects with stability over months in some patients and missed control of disease activity in others who showed relapse activity [25]. In a retrospective multicenter study in NMOSD, there were no beneficial effects of GA treatment regarding the prevention of relapse activity or disability progression, with 14/16 patients having at least one relapse within six months under GA therapy [26]. Another retrospective study also showed no effect in reducing the risk of NMOSD attacks [27]. It thus remains speculative whether GA might be a putative option in NMOSD. 

There are several limitations of the current study which need to be discussed. First, the OSE model represents a crossover model mimicking features of both neuromyelitis optica and MS [9], therefore, not featuring all aspects of the different diseases. Moreover, the mice investigated in our study showed a higher incidence and disease activity than expected and reported in previous studies [8,9,13]. Of note, OSE incidence and the severity of disease progression was higher in untreated male OSE mice, whereas the opposite was observed in GA-treated mice with, however, smaller overall differences between females and males. A higher accumulation of disability in untreated male OSE mice reflects well the situation both in MS and NMOSD, with males being at a higher risk of neurodegeneration and disability progression [28]

While we could document moderate but significant clinical effects, we did not observe any significant histological alterations regarding general infiltration and demyelination. The effects on microglia/macrophages documented in this study have also been reported by others but are presumably not only responsible for the beneficial effects observed in the experiments presented here. 

## 5. Conclusions

We show modest positive effects of prophylactic GA treatment in a spontaneous model of chronic neuroinflammation, mostly mediated by effects on microglia/macrophages and dendritic cells. As GA is a well-known and well-defined therapeutic option, it may represent a potential add-on medication in chronic inflammatory conditions of the CNS. As early therapeutic intervention in chronic CNS inflammation is established as a therapeutic concept to protect the neuronal reserve, this study supports that trying medications with very good safety profiles in the earliest stage of these diseases can be justified.

## Figures and Tables

**Figure 1 cells-12-00542-f001:**
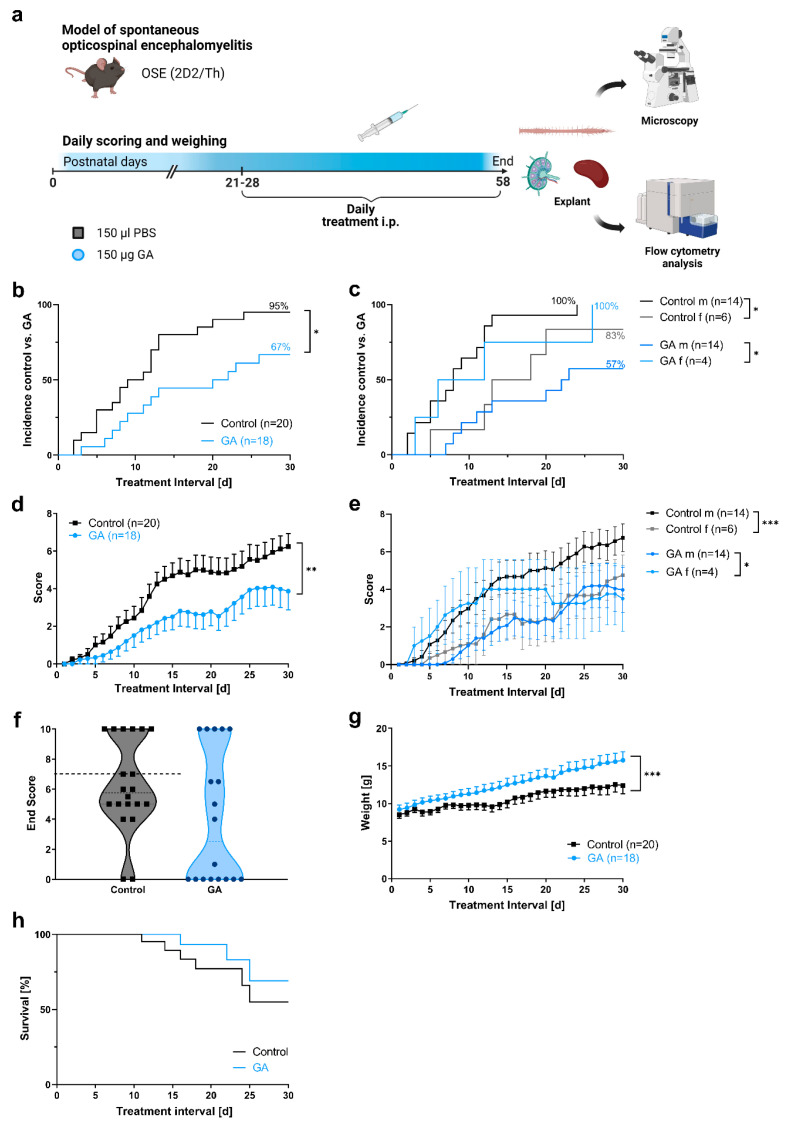
GA positively attenuates disease progression in OSE without effect on survival. (**a**) Overview of the experimental layout (created with Biorender.com). Animals were randomly assigned into either the GA treatment group (blue circles) or the PBS control group (black squares). (**b**) OSE incidence was higher in the control group compared to GA-treated OSE mice (*p* < 0.05). (**c**) Male control mice had a higher incidence of OSE compared to female mice (*p* < 0.05), whereas in female mice, the incidence was higher in GA-treated mice compared to controls (*p* < 0.05). (**d**) GA treatment positively attenuated the clinical signs as measured using the EAE score (*p* < 0.01). (**e**) In the control group, males had a higher score than females (*p* < 0.001). In the GA group, the scores of female mice were higher (*p* < 0.05). (**f**) The overall lower disability under treatment was reflected in a 2.3 point lower end score of GA-treated animals after 30 days (mean score of 3.9 in the GA group versus mean score of 6.2 in the control group) and more animals with a score of 0 in the GA group (n = 8/18) compared to the control group (n = 2/20). The interrupted line represents the cut-off following which mice were sacrificed due to animal care guidelines. (**g**) GA-treated animals had a higher body weight than control mice (*p* < 0.001). (**h**) General survival did not differ significantly between both groups. Control group n = 20; GA n = 18. Data were analyzed using non-parametric two-tailed Mann–Whitney U test and are expressed as mean ± SEM; * *p* < 0.05; ** *p* < 0.01; *** *p* < 0.001.

**Figure 2 cells-12-00542-f002:**
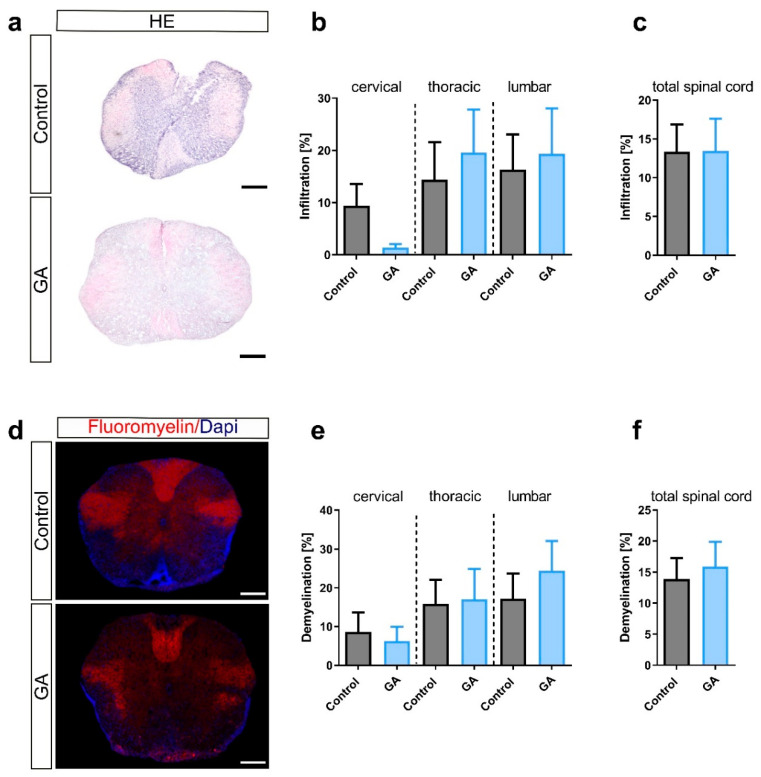
GA treatment has no effect on infiltration and demyelination in the spinal cord. (**a**–**d**) Infiltration was detected via HE staining, (**d**–**f**) demyelination via Fluoromyelin Red. Control mice had infiltration in 13.4% of the white matter of the spinal cord, as shown in representative transversal spinal cord sections. (**a**) Cervical, thoracic and lumbar sections of the spinal cord were analyzed separately and showed no changes of general infiltration upon GA treatment (**b**) as well as analysis of all segments combined (**c**). (**d**) Representative images of Fluromyelin-stained spinal cord cryosections. Accordingly, demyelination did not differ neither in cervical, thoracic and lumbar sections (**e**) nor upon analysis of all segments in combination (**f**). Data were analyzed using one-way ANOVA (**b**,**e**) or non-parametric Kruskal–Wallis test (**c**,**f**). Data are shown as mean ± SEM. Scale bars depict 50 µm.

**Figure 3 cells-12-00542-f003:**
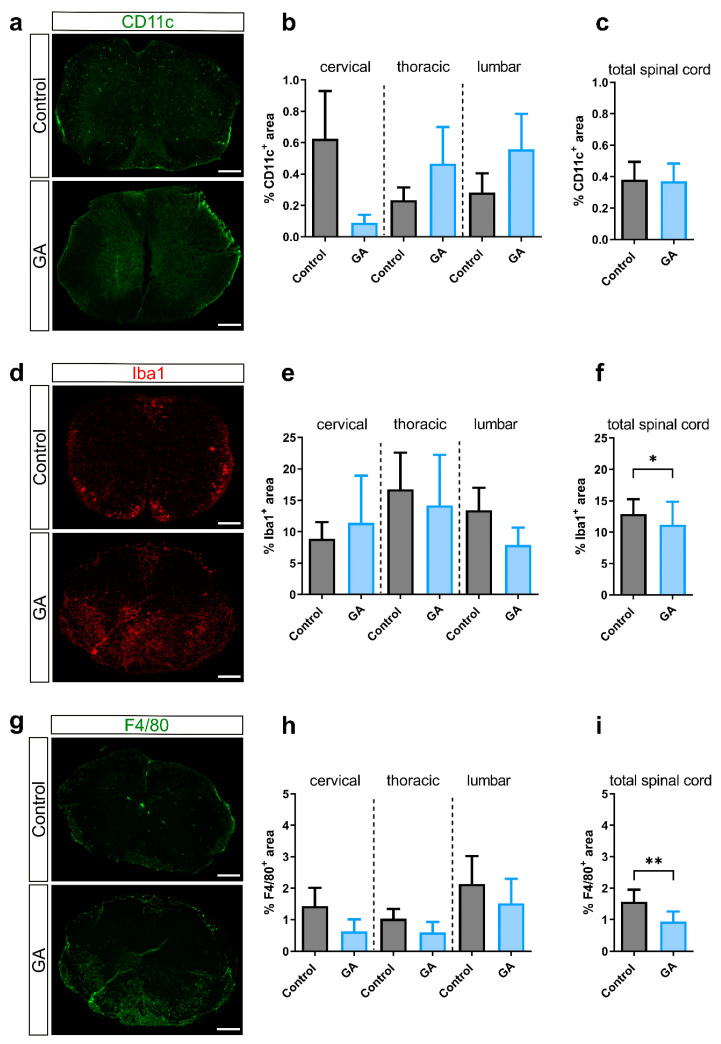
GA alters Iba1^+^ and F4/80^+^ infiltration in OSE. Immunohistochemistry of spinal cord sections of control and GA OSE mice. (**a**) CD11c^+^ staining was performed to investigate dendritic cells. (**b**) In GA-treated animals, CD11c^+^ signal appeared mainly in the thoracolumbar region, which did not reach significance (**b**) without differences in the distribution regarding the total spinal cord (**c**). (**d**) Staining of microglia/macrophages (Iba1, red). (**e**) Quantification showed no significant differences of cervical, thoracic or lumbar cord. (**f**) There was, however, a reduction of Iba1^+^ area upon GA treatment after analysis of all areas of the spinal cord. (**g**) Staining for F4/80^+^ representing activated microglia/macrophages showed a decrease in all areas combined in the GA-treated group (**i**). Data were analyzed using one-way ANOVA (**b**,**e**,**h**) or non-parametric Kruskal–Wallis test (**c**,**f**,**i**). Data are shown as mean ± SEM; * *p* < 0.05; ** *p* < 0.01. Scale bars depict 50 µm.

**Figure 4 cells-12-00542-f004:**
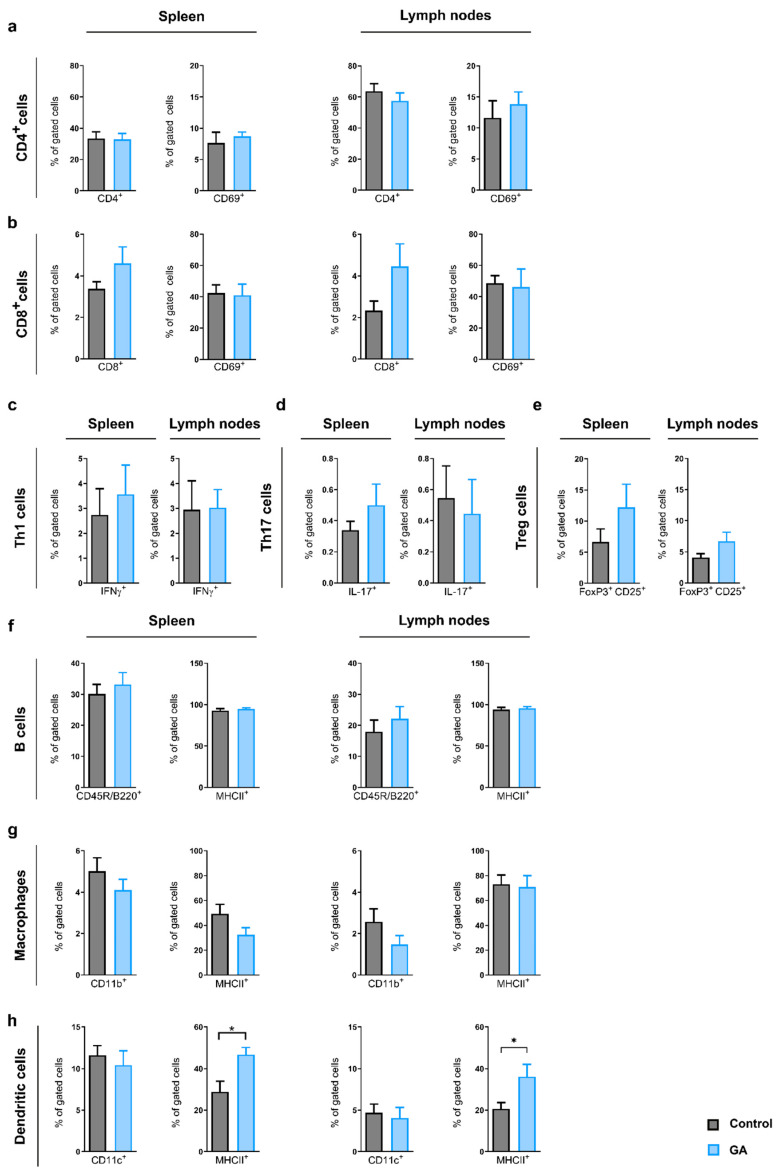
GA treatment enhances frequency of activated dendritic cells without effect on other immune cell subpopulations in secondary lymphoid organs. Frequencies of immune cells in secondary lymphoid organs were analyzed using flow cytometry. (**a**) Frequencies of CD4^+^ T helper (**a**) and CD8^+^ cytotoxic T cells (**b**), as well as activated forms (CD4^+^CD69^+^ and CD8^+^CD69^+^), did not differ between control and GA-treated mice. Th1 and Th17 subpopulations were unaffected (**c**,**d**), while regulatory T cells trended towards higher frequencies in GA-treated mice (**e**). In the spleen and the lymph nodes, frequencies for CD45R/B220^+^ B cells, as well as the activated form (CD45R/B220^+^ MHCII^+^), were unaltered (**f**). In the spleen and the lymph nodes unactivated macrophages (**g**), as well as DC frequencies for both groups, revealed similar findings (h). While activated (CD11b^+^ MHCII^+^) macrophages (**g**) were unaffected, activated (CD11c^+^ MHCII^+^) DCs (**h**) were significantly increased in the GA-treated mice compared to the control group. Data are shown as mean ± SEM; * *p* < 0.05. (Spleen: control n = 11, GA n = 8; lymph nodes: control n = 11, GA n = 9).

**Table 1 cells-12-00542-t001:** OSE incidence and treatment-related scores.

	Animal Sex	Disease Incidence (%)	Incidence Depending on Sex	Mean Score ± SEM	Max Score ± SEM	Mean Day of Onset ± SEM *	Survival (%)
GA	m 14 f 4	12/18 (67)	m 57% f 100%	2.2 ± 0.57	4.4 ± 0.94	13.3 ± 2.19	13/18 (72.2)
Control	m 14 f 6	19/20 (95.0)	m 100% f 83%	3.7 ± 0.51	6.5 ± 0.62	9.9 ± 1.39	14/20 (70.0)

* of diseased animals.

## Data Availability

All data are available from the corresponding author S.F. upon reasonable request.

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
