# Peer review of "Prophylactic Glatiramer Acetate Treatment Positively Attenuates Spontaneous Opticospinal Encephalomyelitis"

_cells, 2023, doi:10.3390/cells12040542_

Round 1

Reviewer 1 Report

careful study in an well-established animal model opticospinal encephalomyelitis (OSE) with resemblance to clinical human diseases multiple sclerosis (MS) and neuromyelitis optica spectrum disorder (NMOSD) using glatiramer acetate (GA). Convincing results re slowing clinical progression in comparison to adequate placebo treatment. Thoughtful discussion of potential mechanisms of these treatments. Applicability of this treatment to clinical human disease still in question. Particularly well written

Reviewer 2 Report

This study of GA effects on OSE mice addressed the question of a potential benefit of this product for MS and NMOSD.  

The presented study on the OSE model is well designed and presents positive results on a global disease scale, which is confirmed by the correlated weight gain curve (better with GA). This is the main and the only significant result that the study has clearly shown, therefore at the clinical level for an animal model that, whatever its common points with the human diseases, cannot predict similar effects of a given product in the real disease as now well experienced with many models.

At the biological level, no difference was found in lymphoid infiltration in GA treated and control mice, but a low significance is reported on innate immune cells (resident microglia and perivascular/infiltrated macrophages if ever and not microglia). This is only seen when pooling the results from all sections and spinal tissue regions. Other differences in peripheral lymphocyte subpopulations are not significant despite a possible trend in GA-induced CD8+ cells deemed to be regulatory T-cells.

Thus, the present study only shows global clinical differences and very mild influence on microglia activation that is known to drive what is called "neurodegenerescence" by neurologists involved in MS studies and "neuroinflammation" by others, e.g., involved in AD or PD studies. Anyhow, this does not predict a benefit for the major factor driving disability and progression in MS.

In conclusion, the animal study by itself is well conducted and discussed, in particular for divergent findings, compared to previous studies, which may be due to different colonies of these transgenic mice.

The translation to human disease is also well discussed and only suggesting that GA treatment is worth "trying" in very early MS and, why not also NMOSD, cases, given its safety profile.

The paper is not providing major information but the study confirms that "trying" products with very good safety profiles in the earliest stage of these diseases can be justified.  This could be better outlined in the concluion.

Reviewer 3 Report

The manuscript by Koc et al. describes the modest positive effects of prophylactic GA treatment in a spontaneous model of chronic neuroinflammation, mostly mediated by effects on microglia/macrophages and dendritic cells. In this way, GA may represent a potential add-on medication in chronic inflammatory conditions of the CNS.

The manuscript presents interesting results and as the authors say, there are several limitations in the current study, which need to be discussed.

11- Mice investigated in their study showed a higher incidence and disease activity than expected and reported in previous studies. The authors should indicate the sex of the animals used, and it would be interesting to analyze the sex and the incidence because maybe the incidence depends on the sex as happens in humans with MS an NMOSD.

22- Another point to discuss is the effects on microglia/macrophages. The authors analyzed the quantity of microglia/macrophages and their activation by immunofluorescence with Iba1 and F4/80 antibodies. First, the author should describe with more details the method how they realize the quantification. Second, it would be interesting to do a double check of the microglia/macrophage activation. The authors might measure the Iba1 immunostaining area of both the microglia cell body and the whole cell to determine microglial morphological changes. The method is described in doi: 10.3390/ijms23105347 and maybe they could use the images than they already have.
